# Understanding the Radiobiological Mechanisms Induced by ^177^Lu-DOTATATE in Comparison to External Beam Radiation Therapy

**DOI:** 10.3390/ijms232012369

**Published:** 2022-10-15

**Authors:** Wendy Delbart, Jirair Karabet, Gwennaëlle Marin, Sébastien Penninckx, Jonathan Derrien, Ghanem E. Ghanem, Patrick Flamen, Zéna Wimana

**Affiliations:** 1Nuclear Medicine Department, Institut Jules Bordet, Université Libre de Bruxelles (ULB), 1070 Brussels, Belgium; 2Laboratory of Oncology and Experimental Surgery, Institut Jules Bordet, Université Libre de Bruxelles (ULB), 1070 Brussels, Belgium; 3Medical Physics Department, Institut Jules Bordet, Université Libre de Bruxelles (ULB), 1070 Brussels, Belgium; 4Laboratoire de Physique Nucléaire et Des Radiations, Institut Supérieur Industriel de Bruxelles (ISIB), 1000 Brussels, Belgium; 5NEMP Applied Research Lab, Institut de Recherche de l’Institut Supérieur Industriel de Bruxelles (IRISIB), 1000 Brussels, Belgium

**Keywords:** radionuclide therapy, ^177^Lu-DOTATATE, external beam radiation therapy, radiobiology, DNA damage, PARP inhibition

## Abstract

Radionuclide Therapy (RNT) with ^177^Lu-DOTATATE targeting somatostatin receptors (SSTRs) in neuroendocrine tumours (NET) has been successfully used in routine clinical practice, mainly leading to stable disease. Radiobiology holds promise for RNT improvement but is often extrapolated from external beam radiation therapy (EBRT) studies despite differences in these two radiation-based treatment modalities. In a panel of six human cancer cell lines expressing SSTRs, common radiobiological endpoints (i.e., cell survival, cell cycle, cell death, oxidative stress and DNA damage) were evaluated over time in ^177^Lu-DOTATATE- and EBRT-treated cells, as well as the radiosensitizing potential of poly (ADP-ribose) polymerase inhibition (PARPi). Our study showed that common radiobiological mechanisms were induced by both ^177^Lu-DOTATATE and EBRT, but to a different extent and/or with variable kinetics, including in the DNA damage response. A higher radiosensitizing potential of PARPi was observed for EBRT compared to ^177^Lu-DOTATATE. Our data reinforce the need for dedicated RNT radiobiology studies, in order to derive its maximum therapeutic benefit.

## 1. Introduction

Targeted Radionuclide Therapy (RNT) is an effective systemic treatment modality relying on the targeted and simultaneous irradiation of multiple diseased/metastatic sites, with limited healthy tissues toxicities. Peptide Receptor Radionuclide Therapy (PRRT) with ^177^Lu-DOTATATE targeting somatostatin receptor sub-type 2 (SSTR2) in neuroendocrine tumours (NET) is one of the most successful developments in nuclear medicine, and currently widely used in clinical routine. However, complete responses remain rare, with stable disease being the main response pattern [1,2,3]. Although no predictive biomarkers have been clinically validated, a combination of prognostic information such as grade or extent of NET and tumour radiosensitivity are known to determine patient outcome after PRRT [4]. Hence, greater radiobiology knowledge may aid in treatment optimization to better select patients as well as increase the efficacy [5,6].

Most of the radiobiology knowledge has been derived from external beam radiation therapy (EBRT) studies, and all paradigms may not be applied to RNT due to the differences related to the nature of the radiation (β^−^ particle for Lu-177 versus electromagnetic photons (X-rays)), the total dose and dose rate (low versus high), the dose distribution pattern (heterogeneous versus homogeneous) and intervals between fractions (8 to 12 weeks apart versus daily fractions) [7]. Identification of the biological differences has not been comprehensively investigated and their possible implication in the clinic warrants to be explored. Such data are needed to fully exploit RNT potential, including in the context of treatment combination strategies. For example, DNA repair targeting strategies (i.e., using poly (ADP-ribose) polymerase (PARP) inhibitors) have shown promise in combination with EBRT and were therefore also tested in combination with RNT; however, different radiobiological considerations may need to be applied for efficient radiosensitization [8].

In this in vitro study, we aimed at evaluating common radiobiological endpoints (i.e., cell survival, cell cycle, cell death, oxidative stress, DNA damage) in ^177^Lu-DOTATATE and EBRT-treated cells over time in a panel of six human cancer cell lines expressing SSTRs and comparing the radiosensitizing potential of PARP inhibition (PARPi).

## 2. Results

### 2.1. Intrinsic Radiosensitivity to ^177^Lu-DOTATATE and EBRT

The radiosensitivity to ^177^Lu-DOTATATE and EBRT of the panel of human cancer cell lines (melanoma, multiple myeloma and gastroenteropancreatic (GEP) carcinoma) was evaluated, using a previously published treatment schedule for ^177^Lu-DOTATATE [9,10] and the clinically applied 2 Gy dose of EBRT [11,12]. The effect of ^177^Lu-DOTATATE on the survival of the six human cancer cell lines expressing SSTRs as a function of time was described in our previous work [9]. The multiple myeloma cell lines COLO-677 and EJM and the melanoma cell line HBL were the most radiosensitive, with a significant cell survival reduction at day 10 of 33% ± 2%, 22% ± 2% and 26% ± 4%, respectively (all *p* values < 0.001). The melanoma cell line MM162 and the GEP cell line MIA-PACA-2 had an intermediate radiosensitivity (cell survival reduction of 13% ± 3% and 14% ± 3% respectively, all *p* values < 0.001), while the survival of the GEP cell line HT-29 was not affected by ^177^Lu-DOTATATE treatment.

A time-dependent decrease in cell survival was also induced after 2 Gy of EBRT (Figure 1a). At day 10, cell survival was 42% ± 2% (COLO-677), 45% ± 4% (EJM), 51% ± 3% (MIA-PACA-2), 61% ± 3% (HBL), 62% ± 6% (MM162) and 86% ± 6% (HT-29) of the non-treated counterpart (Figure 1b). HT-29 was described as a radioresistant cell line by other groups [13,14,15]. Cell survival reduction was significantly more pronounced in all cell lines compared to ^177^Lu-DOTATATE (all *p* values < 0.001).

Ranking cell lines according to their ^177^Lu-DOTATATE radiosensitivity did not perfectly match EBRT radiosensitivity (Pearson coefficient r = 0.81, *p* = 0.05). The melanoma HBL cell line was as sensitive as multiple myeloma cell lines to ^177^Lu-DOTATATE, but not EBRT (Figure 1b).

### 2.2. Effect of ^177^Lu-DOTATATE and EBRT on Cell Cycle

In order to assess if the cell survival decrease following irradiation with ^177^Lu-DOTATATE and EBRT was (partially) due to a decrease in cell proliferation (cytostatic effect), the cell cycle distribution was assessed at a late time point (10 days). Neither ^177^Lu-DOTATATE nor EBRT changed the cell cycle distribution compared to the non-treated counterpart (Figure 2, Appendix A).

### 2.3. Effect of ^177^Lu-DOTATATE and EBRT on Apoptosis

Apoptotic levels in the control samples increased with time in COLO-677, MIA-PACA-2 and HT-29 cell lines.

Apoptosis was significantly induced by ^177^Lu-DOTATATE only at day 10 in the three ^177^Lu-DOTATATE-sensitive cell lines compared to the control: +70% ± 21% (*p* = 0.015) in HBL, +27% ± 9% (*p* = 0.001) in COLO-677 and +35% ± 12% (*p* = 0.031) in EJM. On the contrary, the EBRT-sensitive cell lines had already significant increased apoptotic levels at day 3 after EBRT: +52% ± 13% (*p* = 0.009) in COLO-677, +42% ± 14% (*p* = 0.017) in EJM and +55% ± 15% (*p* = 0.017) in MIA-PACA-2. This increase was maintained at day 10: +42% ± 12% (*p* = 0.001) in COLO-677, +71% ± 17% (*p* < 0.001) in EJM and +50% ± 9% (*p* < 0.001) in MIA-PACA-2. Apoptotic levels were significantly lower after ^177^Lu-DOTATATE compared to EBRT, except in the HBL and MM162 melanoma cell lines (Figure 3).

### 2.4. Effect of ^177^Lu-DOTATATE and EBRT on Autophagy

In the ^177^Lu-DOTATATE-sensitive cell lines, increased autophagy was already observed at day 3 and further amplified at day 10 after ^177^Lu-DOTATATE compared to the non-treated counterpart in HBL (D3: +4.3% ± 0.8%, *p* < 0.001; D10: +7.8% ± 2.0%, *p* < 0.001) and EJM (D3: +6.4% ± 2.3%, *p* = 0.013; D10: +15.1% ± 2.5%, *p* < 0.001), except for COLO-677 that increased only at the later time point (D10: +17.3% ± 1.5%, *p* < 0.001). In contrast, the less ^177^Lu-DOTATATE-sensitive cell lines also demonstrated increased autophagy, however, a return to basal levels was observed in MM162 (D3: +12.9% ± 3.5%, *p* = 0.004; D10: +1.4% ± 4.6%) and MIA-PACA-2 (D3: +8.7% ± 1.5%, *p* = 0.006; D10: +0.4% ± 3.4%). Although autophagy induction was significantly higher after EBRT compared to ^177^Lu-DOTATATE in all cell lines but one (MM162), the same time dependency could be observed with further amplification at day 10 in HBL (D3: +11.7% ± 0.2%, *p* < 0.001; D10: +17.6% ± 2.9%, *p* < 0.001), COLO-677 (D3: +15.7% ± 3.0%, *p* = 0.001; D10: +32.8% ± 5.4%, *p* < 0.001), EJM (D3: +24.0% ± 4.9%, *p* = 0.001; D10: +30.5% ± 6.5%, *p* < 0.001), or reversion at day 10 in MM162 (D3: +31.5% ± 4.9%, *p* = 0.002; D10: +21.4% ± 13.8%, *p* = 0.03) and in MIA-PACA-2 (D3: +23.05% ± 3.6%, *p* < 0.001; D10: −5.9% ± 5.9%). On the other hand, in the HT-29 cell line, no autophagy induction was observed at any of the investigated time points neither after ^177^Lu-DOTATATE nor EBRT (Figure 4).

### 2.5. Effect of ^177^Lu-DOTATATE and EBRT on Reactive Oxygen Species (ROS)

After ^177^Lu-DOTATATE exposure, ROS levels were significantly increased early on (15 min) in MM162 (+7% ± 1%, *p* = 0.005) and COLO-677 (+17% ± 4%, *p* = 0.04), however, this elevated status was not statistically significant at day 3 (MM162: +8% ± 3%, *p* = 0.08; COLO-677: +14% ± 5%, *p* = 0.06), where only in MIA-PACA-2 (+16% ± 4%, *p* = 0.02) a significant later increase in ROS levels was observed. In contrast, EBRT induced a significant ROS level increase early in most cell lines (MM162: +34% ± 6%, *p* = 0.005; COLO-677: +46% ± 2%, *p* = 0.001; EJM: +24% ± 4%, *p* < 0.001; MIA-PACA-2: +18% ± 2%, *p* < 0.001), which was further enhanced at day 3 in MM162: +52% ± 12%, *p* = 0.01; COLO-677: +57% ± 8%, *p* = 0.003 and MIA-PACA-2: +45% ± 5%, *p* = 0.001. In comparison to ^177^Lu-DOTATATE, ROS levels observed after EBRT were higher in MM162, COLO-677, EJM and MIA-PACA-2 within 15 min after irradiation, and in MM162, COLO-677 and MIA-PACA-2 at day 3. Once more, in the HT-29 cell line, no ROS level increase was observed at any of the investigated time points neither after ^177^Lu-DOTATATE nor EBRT (Figure 5).

### 2.6. Effect of ^177^Lu-DOTATATE and EBRT on DNA Damage

DNA double-strand breaks (DSBs) were assessed in ^177^Lu-DOTATATE- and EBRT-treated cells until 10 days post irradiation using γH2AX and pATM as early DNA damage response markers. γH2AX/pATM levels of the control were quite stable over time and in a similar range in all cell lines: HBL (5.87–9.45%), MM162 (8.68–12.72%), COLO-677 (9.90–14.45%), EJM (10.37–17.22%), MIA-PACA-2 (10.25–16.92%) and HT-29 (5.01–8.98%). EBRT exposure led to a marked early peak increase in γH2AX/pATM levels in all cell lines around 15 min to 1 h post irradiation (depending on the cell lines) (HBL: 4.5-fold increase; MM162: 3.2-fold increase; COLO-677: 3.2-fold increase; EJM: 2.4-fold increase; MIA-PACA-2: 2.9-fold increase; HT-29: 2.9-fold increase). In sharp contrast to EBRT, no clear peak increase in γH2AX/pATM levels could be observed in any cell lines after ^177^Lu-DOTATATE exposure. However, ^177^Lu-DOTATATE exposure led to a smaller but constant increase in γH2AX/pATM levels compared to the control in HBL (1.25–1.38-fold increase), COLO-677 (1.34–1.48-fold increase), EJM (1.35–1.61-fold increase), MIA-PACA-2 (1.18–1.75-fold increase) and HT-29 (1.20–1.52-fold increase). No change from baseline was observed in MM162 over time (Figure 6).

### 2.7. Effect of Olaparib in Combination with ^177^Lu-DOTATATE or EBRT on Cell Survival

Given the difference in DNA damage induction between ^177^Lu-DOTATATE and EBRT, we assessed the radiosensitizing potential of the PARP inhibitor olaparib in combination with ^177^Lu-DOTATATE or EBRT. The varying sensitivities to olaparib among cell lines required the use of different concentrations in order to cause a minimal effect of olaparib as monotherapy on cell survival (90–100% of cell survival compared to the non-treated counterpart) to assess the full radiosensitizing potential of olaparib. COLO-677 was the most sensitive cell line to olaparib, requiring the lowest concentration of 10^−8^ M to achieve a low toxic dose of olaparib. Not surprisingly, COLO-677 had the highest PARP1 expression level among the cell lines (Appendix A), and it was shown that PARP1 expression positively correlated with PARP inhibitor sensitivity [16]. Olaparib further decreased cell survival in combination with both ^177^Lu-DOTATATE and EBRT, however, not in all cell lines. The combination reduced cell survival compared to radiation (^177^Lu-DOTATATE and EBRT) alone in the HBL and MM162 melanoma cell lines, and only in combination with EBRT in the MIA-PACA-2 and HT-29 GEP cell lines. The combination of olaparib and radiation did not affect cell survival of COLO-677 and EJM multiple myeloma cell lines compared to radiation alone (Figure 7) (survival percentages reported in Appendix A). In melanoma and GEP cell lines, the combinations resulted in a synergistic effect, hence radiosensitization, as shown by a coefficient of drug interaction (CDI) < 1. Radiosensitization was more pronounced in combination with EBRT with a lower CDI and a larger amplification factor (AF) compared to ^177^Lu-DOTATATE (Table 1).

### 2.8. Absorbed Dose Calculation of ^177^Lu-DOTATATE Treatment

The ^177^Lu-DOTATATE experimental treatment scheme (4 h incubation with 5 MBq ^177^Lu-DOTATATE followed by an incubation period of 10 days) was estimated to deliver an absorbed dose to cells of 4.2 ± 0.2 Gy (sphere model) and 4.4 ± 0.3 Gy (semi-ellipse model). The major contributor to the total absorbed dose was the radioactive medium during the 4 h incubation: 84.9% ± 6.1% for the sphere model and 88.2% ± 9.6% for the semi-ellipse model. The contribution of the internalized fraction during the 10-day incubation was 14.8% ± 0.8% (sphere model) and 11.6% ± 0.9% (semi-ellipse model) (Figure 8).

## 3. Discussion

Radiobiology knowledge mostly stems from EBRT studies and is often extrapolated to RNT. However, it is becoming accepted that dedicated RNT radiobiology studies should be undertaken [5,6], since these two radiation modalities are characterized by distinct radiation features. Indeed, the protracted exposure to a low dose rate (exponentially decreasing according to the half-life of the therapeutic radioisotope) and the heterogeneous dose distribution pattern (due to the heterogeneous distribution of radioactivity in cells and nature of particle tracks) are inherent characteristics of RNT that can lead to different radiobiological responses compared to EBRT [17]. In this in vitro study, common radiobiological endpoints, i.e., cell survival, cell cycle, cell death, oxidative stress and DNA damage, were assessed over time after ^177^Lu-DOTATATE and EBRT in a panel of human cancer cell lines expressing SSTRs. Furthermore, the radiosensitizing potential of the PARP inhibitor olaparib in both treatments was investigated.

We showed that common radiobiological mechanisms were induced by both ^177^Lu-DOTATATE (5 MBq corresponding to 4 Gy) and EBRT (2 Gy), nevertheless to a different extent and/or with variable kinetics.

EBRT was more efficient in decreasing cell survival than ^177^Lu-DOTATATE in all cell lines independent of their type. Interestingly, the relative ^177^Lu-DOTATATE radiosensitivity among cell lines did not perfectly match EBRT radiosensitivity, suggesting potential differences in the underlying radiobiological mechanisms. This was also shown by O’Neill E. et al., who reported non-linear correlation of radiosensitivity between ^177^Lu-DOTATATE and EBRT in SSTR-expressing cell lines [18]. Although the crystal violet assay used in this study might have yielded a higher cell survival fraction compared to the EBRT gold standard clonogenic assay, the relative radiosensitivity among cell lines was not expected to be affected by the assay choice. Indeed, crystal violet was shown to be an adequate method to compare the relative radiosensitivities of different cell lines [19].

A cytostatic effect of either ^177^Lu-DOTATATE or EBRT, that could (partially) explain the decreased survival fraction of cells, could not be demonstrated in our experiment. In contrast, their cytotoxic nature was observed. Apoptosis was already induced at day 3 after EBRT in EBRT-sensitive cell lines (COLO-677, EJM, MIA-PACA-2), but only at day 10 in the ^177^Lu-DOTATATE-sensitive ones (COLO-677, EJM, HBL). As for autophagy, the kinetic profile was similar between ^177^Lu-DOTATATE and EBRT, but the level of induction was lower in ^177^Lu-DOTATATE-treated cells. Part of this cytotoxic effect of ^177^Lu-DOTATATE and EBRT involves ionizing radiation (IR) indirect effects and ROS. As expected, ROS increased within 15 min after EBRT and also at day 3, which may represent delayed ROS caused by mitochondrial metabolism after irradiation [20,21]. Surprisingly, only two cell lines had elevated ROS levels after 4 h incubation with ^177^Lu-DOTATATE and one cell line at day 3. Yet, ROS-mediated indirect effects by low LET radiation such as β- particles emitted by Lu-177 should account for about two-thirds of the biological effects induced. The low dose rate together with the very transient nature of ROS may explain the surprisingly low to non-detectable levels of total cellular ROS in ^177^Lu-DOTATATE-treated cells, in contrast to EBRT-treated cells. The assessment of more stable markers of oxidative stress compared to ROS such as lipid peroxides or protein carbonyl content [22] may bring additional information on the role of ROS in RNT outcome.

The big paradigm in radiobiology relies on DNA DSBs as the main event leading to IR-induced clonogenic cell death. In contrast to EBRT, no H2AX/ATM phosphorylation peak was visible after ^177^Lu-DOTATATE treatment in any of the cell lines in a 10-day time frame, but rather a slightly higher proportion of γH2AX/pATM-positive cells compared to the non-treated counterpart that tended to be stable in time in some cell lines. The difference in the kinetics of the DNA damage response (DDR) as well as the level of induction between ^177^Lu-DOTATATE and EBRT was also highlighted by O’Neill et al. [18], showing a high induction of γH2AX foci early after EBRT, as opposed to a lower but prolonged induction after ^177^Lu-DOTATATE in the rat pancreatic cancer cell line CA20948. While the high dose rate used in EBRT intends to overwhelm the DNA repair capacity, the low dose rate applied with ^177^Lu-DOTATATE in a protracted exposure gives cells sufficient time to repair a fraction of damage before the creation of subsequent breaks. Consequently, DNA DSB formation is competing with DNA damage repair in this time window [23], resulting in a challenging distinction from background variability. Additional differences related to distribution pattern as well as size of foci, representing DNA damage complexity, might be expected between ^177^Lu-DOTATATE and EBRT and could be addressed by looking at individual cells by radiation-induced foci detection by microscopy. Nevertheless, flow cytometry analysis allowed us to semi-quantitatively assess a large number of conditions (cell lines and time points) to have a first general idea on the extent and kinetics differences between ^177^Lu-DOTATATE and EBRT in a panel of cancer cell lines. In EBRT-treated cells, elevated γH2AX/pATM levels observed from 72 h post irradiation might represent a combination of residual unrepaired DSBs, but also apoptotic cells which have also been associated with high γH2AX expression [23,24]. Indeed, apoptosis was observed to be already induced at day 3 in our cell lines, namely COLO-677, EJM and MIA-PACA-2 after EBRT.

Given the sharp differences in DNA damage induction between ^177^Lu-DOTATATE and EBRT-treated cells, we assessed the radiosensitizing potential of the PARP inhibitor olaparib in combination with ^177^Lu-DOTATATE or EBRT. Olaparib radiosensitized cells to both ^177^Lu-DOTATATE and EBRT, although not in all cell lines, and the synergistic effect was more pronounced in combination with EBRT (lower CDI and larger AF) compared to ^177^Lu-DOTATATE. This radiosensitizing effect was previously shown by other groups with EBRT [25,26] and in SSTR-expressing xenograft mouse models using talazoparib [27] or fluzoparib [28] in combination with ^177^Lu-DOTATATE. Nonetheless, the greater amount of DNA lesions generated by the higher dose rate of EBRT compared to ^177^Lu-DOTATATE may lead to a higher number of cytotoxic DSBs following PARP inhibition (PARPi), and therefore may explain the higher radiosensitizing potential.

The rationale of PARPi-mediated radiosensitization is that IR induces single-strand Breaks (SSBs) that are left unrepaired in cells in which PARP is inhibited, converting them to lethal DSBs at replication [29]. PARP activity biomarkers (i.e., PARylation) after IR may more accurately predict PARPi-mediated radiosensitization and may aid in optimal schedule finding of PARP inhibitors in combination with IR. Indeed, the full radiosensitizing potential of olaparib is likely to be achieved using distinct schedules and doses. For ^177^Lu-DOTATATE, a dosimetry-based schedule rationale is to start the PARP inhibitor 24 h after the infusion of ^177^Lu-DOTATATATE and continue for 4 weeks at each cycle [8]. As for EBRT, Verhagen et al. suggested that a short 7 h exposure (1 h before and 6 h after irradiation) of the PARP inhibitor is sufficient for radiosensitization [30]. The minimum inhibitory dose (optimum biological dose) may also differ considering the differences in DNA damage induced, the different organs at risk and the potential DDR genetic background differences in cancer types treated by ^177^Lu-DOTATATE and EBRT. Results of ongoing clinical trials investigating the combination of PARP inhibitors and ^177^Lu-DOTATATE are awaited (NCT05053854, NCT04375267, NCT04086485).

The absence of radiosensitization in some cell lines may be explained by resistance mechanisms to PARPi [31], insufficient IR-mediated SSB generation to lead to a synergistic effect or possibly non-effective PARP inhibitor concentrations. In our study, olaparib concentrations given on a continuous basis were chosen to have a minimal cytotoxic effect at the time of survival assessment (day 10). Shorter exposure with higher concentrations might have been more beneficial for efficient radiosensitization. Nevertheless, in patients, the dose to achieve efficient radiosensitization must always be balanced against drug toxicities. The absence of ^177^Lu-labelled Prostate-Specific Membrane Antigen (PSMA) radiosensitization by different PARP inhibitors in prostate cancer cell lines was reported in a recent study [32]. Additional studies are needed to decipher PARPi optimal conditions for efficient RNT radiosensitization.

The absorbed dose delivered to the cells by 5 MBq of ^177^Lu-DOTATATE was estimated to be in the range of 4.2–4.4 Gy, a two times higher absorbed dose compared to EBRT. However, the ^177^Lu-DOTATATE-induced biological effects were still lower, indicating a lower in vitro relative biological effectiveness (RBE). The dose rate most likely accounts for the major parameter involved in the radiation-induced biological response differences [33]. For EBRT, 2 Gy was delivered within a minute, while 4 h exposure to ^177^Lu-DOTATATE was needed to achieve 4 Gy (corresponding to about 16 mGy/min). Gholami et al. already reported a lower radiobiological effectiveness of the Y-90 low-dose rate compared to Y-90 high-dose rate and EBRT in colorectal cell lines [14]. The modest biological effects reported in ^177^Lu-DOTATATE-treated cells may also be due to the low internalized fraction in our study. Conversely, the U2OS^SSTR2^, an SSTR-negative human osteosarcoma cell line transfected to stably express SSTR2 [34], had a 250 times higher internalized fraction (250 mBq/cell) compared to our HBL cell line (1 mBq/cell in [9]) after 2 h incubation with ^177^Lu-DOTATATE (in both cases). This resulted in a minor contribution of the internalized fraction to the total absorbed dose. Nevertheless, the biological effects induced either from the radioactive medium or the internalized activity in the final cellular outcome cannot be dissociated in our experiments. Hence, due to the main contribution of the medium to the absorbed dose, even though the differences in cell-related characteristics (e.g., cell geometry, internalized fraction) were not taken into account in our absorbed dose simulations, no major differences between the different cell lines’ absorbed dose were expected. Moreover, these cell-related characteristics may be more determinant for nucleus-absorbed dose calculation [34] compared to cell-absorbed dose calculation.

Overall, the radiobiological findings from our study have some experimental boundaries that have to be considered, namely our in vitro model, our treatment and our data-sampling scheme. All our experiments were conducted in a simplified 2D model that lacks the crossfire effect found in a 3D situation. Additionally, the influence of the microenvironment was also completely absent, although this has been reported to have an impact on RNT efficacy [33]. On the other hand, our cells were exposed to a single ^177^Lu-DOTATATE treatment compared to four treatment cycles usually experienced by NET in clinical practice. In addition, cells were exposed to a full activity during 4 h, without consideration of the known pharmacokinetic impact of blood clearance (bi-exponential decay). For our data collection, we have only investigated a limited series of time points. However, time response is relevant to consider given the protracted irradiation from ^177^Lu-DOTATATE, and it was shown that over time ^177^Lu-DOTATATE treatment affected different cellular functions [35]. For all the reasons cited above, the radiobiological effects of ^177^Lu-DOTATATE might have been underestimated in our study. Nonetheless, our study can serve as a direction to future studies that could be further extended on those subjects.

In conclusion, we highlighted that common radiobiological mechanisms were induced by both ^177^Lu-DOTATATE and EBRT, however, differences mainly lied in the kinetics and the extent of the responses induced, which were fainter for PRRT—even at a higher dose compared to EBRT—and therefore highlight the need to be cautious when extrapolating EBRT radiobiology to RNT. The most striking difference relates to DNA damage response with a dissimilar profile between ^177^Lu-DOTATATE and EBRT. These differences may have clinical implications when designing new combination strategies with an EBRT-based rationale. Dedicated RNT radiobiological studies are needed for optimization of therapeutic radiopharmaceuticals including the use of rationally designed combination strategies.

## 4. Materials and Methods

### 4.1. Cell Lines and Cell Culture

The melanoma cell lines (HBL [36,37] and MM162 [38]) were established in our laboratory. Multiple myeloma (COLO-677 and EJM) and GEP (pancreas carcinoma, MIA-PACA-2 and colon adenocarcinoma, HT-29) cell lines were obtained from DSMZ (Braunschweig, Germany). HBL and MM162 were cultured in Ham’s F10 medium (Lonza), COLO-677, HT-29 was cultured in RPMI-1640 (Sigma), EJM was cultured in Iscove’s MDM (Gibco, Invitrogen, UK) and MIA-PACA-2 was cultured in DMEM (Sigma). All media were supplemented with L-glutamine (Sigma), penicillin (Sigma), streptomycin (Gibco, Invitrogen, UK) and kanamycin (Bio Basic) at standard concentrations as well as with 10% foetal bovine serum, except EJM cell line with 20%. Cells were maintained in their respective growth medium at 37 °C in a humidified 95% air and 5% CO2 atmosphere. All cell lines were regularly checked for mycoplasma contamination using MycoAlert^®^ Mycoplasma Detection Kit (Lonza, Rockland, ME, USA). Cell line authentication was performed with a short tandem repeat (STR) test (Eurofins Genomics, Germany). Cell lines were chosen based on two characteristics: (i) cell lines derived from malignancies expressing SSTRs in order to ensure the presence of the target [9] and (ii) cell lines derived from malignancies exhibiting a range of intrinsic radiosensitivities (from radiosensitive (myeloma) to radioresistant (melanoma) and intermediate radiosensitivity (pancreatic and colorectal cancers), classified as such based on the mean survival at 2 Gy of EBRT [39]).

### 4.2. The ^177^Lu-DOTATATE Production

The ^177^Lu-DOTATATE was produced for clinical use within the radiopharmacy facility of the department of nuclear medicine at Institut Jules Bordet, as previously described [40]. Labelling was performed by a fully automated process using a synthesis module with disposable cassettes (Modular Pharmlab, Eckert & Ziegler, Berlin, Germany). An amount of 9 GBq of non-carrier was added ^177^LuCl_3_ (EndolucinBeta^®^, Ph.Eur, ITM, Germany) and 150 µg DOTATATE (Bachem AG) in sodium ascorbate buffer was heated for 20 min at 80 °C. The obtained raw radioactive solution was purified by solid phase extraction on a C18 cartridge. The radiolabelled peptide was then eluted with 1 mL of 50% ethanol, followed by 19 mL of saline and subsequent sterile filtration over a 0.22 µm filter (included in the disposable cassette). All quality controls were performed according to the European Pharmacopoeia, allowing conditional release of the radiopharmaceutical after appearance, pH, radiochemical purity (specification: >95%) and pyrogenicity testing, and subsequent final release after sterility results (after use).

### 4.3. The ^177^Lu-DOTATATE, EBRT and Olaparib Treatments

Cells were seeded in 12-well plates (Corning^®^ CellBIND^®^ Multiple Well Plate, Merck) at different densities according to the time point evaluated (Table A1). The next day (corresponding to day 0), cells were irradiated, either with ^177^Lu-DOTATATE as previously described [9] or with EBRT.

^177^Lu-DOTATATE: 5 MBq was added in each well, in four replicates. After 4 h of incubation at 37 °C, the medium containing ^177^Lu-DOTATATE was removed and replaced with fresh medium.EBRT: cells were irradiated at a dose of 2 Gy with a 6 MV beam from a Clinac 600 linear accelerator (Varian Medical Systems, Palo Alto, CA, USA). The collimator opening was set to 40 × 40 cm^2^, which gave the possibility of irradiating several plates in one batch. In order to achieve a good dose homogeneity and electronic equilibrium, a 6 mm thick polystyrene build-up was put on top of the plates. Plates were placed on a 5 cm thick polystyrene phantom for adequate backscattering conditions. The dose rate was set to 4 Gy per minute. In order to be consistent with ^177^Lu-DOTATATE treatment conditions, medium was replaced right after EBRT.

Three days after radiation-based treatments, the medium was replaced with fresh medium. Cells were kept in culture until used for various assessments.

In the combination experiments, olaparib (Selleckchem) was added in the culture medium the day before ^177^Lu-DOTATATE/EBRT at given concentrations (10^−6^ M for HBL, MM162 and HT-29; 10^−7^ M for MIA-PACA-2 and EJM; 10^−8^ M for COLO-677). Three days after radiation-based treatments, the medium was replaced with fresh medium with or without olaparib. Cells were kept in culture until used for crystal violet assay on day 10. The coefficient of drug interaction (*CDI*) was used to analyse the interactions between ^177^Lu-DOTATATE/EBRT and olaparib.
$$CDI=\frac{survival\%\left(A+B\right)}{survival\%\left(A\right)\times survival\%\left(B\right)}$$
where *A* is either ^177^Lu-DOTATATE or EBRT and *B* is olaparib. *CDI*  <  1 indicates synergism, *CDI*  =  1 additivity and CD  >  1 antagonism between the drug and the radiation.

An amplification factor (*AF*) was calculated to assess the magnitude of the radiosensitization effect induced by olaparib in combination with ^177^Lu-DOTATATE/EBRT.
$$AF(\%)=\frac{survival\%\left(A\right)-survival\%\left(A+B\right)}{survival\%\left(A\right)}\times 100$$
where *A* is either ^177^Lu-DOTATATE or EBRT and *B* is olaparib.

### 4.4. Cell Dosimetry

Dosimetry was performed to calculate the absorbed dose to the cells treated with ^177^Lu-DOTATATE. Monte Carlo simulations were performed with the Geant4 Toolkit version 10.6. Penelope low-energy electromagnetic physics package was used to describe interactions down to an electron cut-off energy of 250 eV [41,42]. This energy cut-off was used to consider the electron’s range in the nm sized organelle.

#### 4.4.1. Geometry Set-Up

A thousand cells were randomly distributed over the surface (3.8 cm^2^) of a well of a 12-well cell culture plate made of polystyrene, filled with 1 mL of water. Simplified cell geometries were simulated: a sphere and a semi-ellipsoid cell model of matching volume (3052 µm^3^ and 3020 µm^3^, respectively). They were composed of a nucleus (12 µm diameter sphere (sphere model) and ellipsoidal nucleus (30% of the cellular volume) (semi-ellipsoid model), density ρ = 1 g/cm^3^) [43], cytoplasm (density ρ = 1 g/cm^3^) and cell membrane (thickness of 5 nm, made of lipids: ρ = 0.92 g/cm^3^) [34] (Figure 9). Semi-ellipsoid cells were constructed using a minor (a) to major (b) axes ratio of 1/3 and a height (h) to minor axis (a) ratio of 1/2, defined arbitrarily (Figure 9b).

#### 4.4.2. Radioactive Source

The Lu-177 radioactive decay was simulated with sources of electrons and photons [44] as described Table 2. Only X-rays and Auger electrons with absolute intensities over 1% were simulated.

The radiation sources were uniformly distributed in both the 1 mL medium covering the cells in the well and in the cell cytoplasm to simulate the irradiation due to internalized fraction of ^177^Lu-DOTATATE. 

#### 4.4.3. Absorbed Dose Calculation

Absorbed dose was calculated using the following formulas describing the number of disintegrations occurring (1) in the medium during the 4 h incubation phase, $N_{\mathrm{med}}\left(t\right)$; (2) in the cells during the 4 h incubation phase, $N_{\mathrm{int}\_4h}\left(t\right)\;$and (3) in the cells during the next 10 days, $N_{\mathrm{int}\_10d}(t)\;$(Appendix A):(1)$$N_{\mathrm{med}}(t)=\int_{0h}^{4h}{(A}_{0}-A_{\mathrm{int}}\left(t\right)){\cdot e}^{-\lambda \cdot t}\;\mathrm{dt}\;\approx \;\int_{0h}^{4h}A_{0}\cdot e^{-\lambda \cdot t}\;dt$$
(2)$$N_{\mathrm{int}\_4h}\left(t\right)=\int_{0h}^{4h}A_{\mathrm{int}}\left(t\right){\cdot e}^{-\lambda \cdot t}\;\mathrm{dt}=\int_{0h}^{4h}\left[A_{\mathrm{int}\_\mathrm{frac}}\cdot \left({1-e}^{-a\cdot t}\right)\right]{\cdot e}^{-\lambda \cdot t}\mathrm{dt}$$
(3)$$N_{\mathrm{int}\_10d}(t)=\int_{0h}^{236h}A_{\mathrm{int}\_\mathrm{frac}}{\cdot \;e}^{-\lambda \cdot t}\;\mathrm{dt}$$
where:

$A_{0}$ corresponds to the initial activity added into the well (5 MBq).

$A_{\mathrm{int}}\left(t\right)\;$is the internalized activity during the 4 h incubation phase.

$A_{\mathrm{int}\_\mathrm{frac}}$ is the maximum value of internalization fraction (plateau), thus, the total activity internalized in the cells at the end of the 4 h incubation phase. The greatest internalized fraction of the different cell lines, being the one from the HBL melanoma cell line and obtained in our previous study [9], was used for the simulations. 

A corresponds to the internalization rate of the radiopharmaceutical in the cell. 

The absorbed dose to the cell was computed for each phase separately (4 h incubation phase followed by a 10-day (236 h) incubation phase). First, the simulated deposited energies [eV] in nucleus, cytoplasm and membrane were summed and then divided by the mass of the cell and the simulated cumulated activity [MBq.s] (number of disintegrations) to calculate the S value [Gy/(MBq.s)]. Finally, the obtained S-value was multiplied by the cumulated activity for each phase. To ensure a statistical error below 5%, 10 simulations of 5 × 10^8^ incident particles were repeated for each of the different phases.

It was assumed that no radioligand was cleared from the cells once it was bound. Cell proliferation, proximity variation and cell death were not taken into account in the simulations. 

### 4.5. Crystal Violet Assay

Crystal violet was used for cell viability quantification. The culture medium was removed, and cells were gently washed with PBS, fixed with 1% glutaraldehyde (Merck) in PBS for 15 min and stained with 0.1% crystal violet (Sigma) in water for 30 min. The plates were washed under running tap water and subsequently lysed with 0.2% Triton X-100 (Roche) in water for 90 min. The associated absorbance was measured at 590 nm using the BioTek^®^ 800™ TS Absorbance Reader.

### 4.6. Cell Cycle Analysis

Supernatant and adherent cells of four wells were harvested and pooled, washed once with PBS and fixed in 70% ethanol at −20 °C for two hours. Cells were then centrifuged for 5 min and washed once with PBS. After another centrifugation step, cells were resuspended in 400 µL of staining solution per million cells (50 μg/mL propidium iodide (ThermoFisher, Waltham, MA, USA) in PBS) and incubated for 10 min in the dark at room temperature. Samples were acquired on a flow cytometer (Navios EX, Beckman Coulter) and results were analysed using Kaluza Flow Cytometry Analysis v2.1 software (Beckman Coulter, Pasadena, CA, USA).

### 4.7. Apoptosis Analysis

The percentage of apoptotic cells was assessed using the Muse^®^ Annexin V & Dead Cell Kit (Luminex), allowing to distinguish four distinct populations: non-apoptotic cells (Annexin V- and 7-AAD-), early apoptotic cells (Annexin V+ and 7-AAD-), late stage apoptotic and dead cells (Annexin V+ and 7-AAD+) and nuclear debris (Annexin V- and 7-AAD+). Briefly, supernatant and adherent cells of four wells were harvested and pooled. Cell suspension was incubated for 20 min in the dark at room temperature with the Muse^®^ Annexin V & Dead Cell Reagent (Luminex). Every sample was then thoroughly mixed and run on the Muse^®^ Cell Analyzer (Merck).

### 4.8. Autophagy Analysis

Autophagic vacuoles were detected using the Autophagy Detection Kit (abcam). Briefly, supernatant and adherent cells of four wells were harvested, pooled, resuspended in a HEPES solution containing the green detection reagent and incubated for 30 min at 37 °C. Samples were acquired on a flow cytometer (Navios EX, Beckman Coulter, Pasadena, CA, USA) and results were analysed using Kaluza Flow Cytometry Analysis v2.1 software (Beckman Coulter, Pasadena, CA, USA). Relative fluorescence intensities were obtained from the median fluorescence intensity of each histogram of the different experimental conditions.

### 4.9. ROS Measurement

ROS levels were analysed using the fluorogenic probe CellROX^®^ Green Reagent (Invitrogen, ThermoFisher Scientific, Waltham, MA, USA). At the specified time points after radiation-based treatments, the CellRox reagent was added in the culture medium to a final concentration of 5 µM. After 30 min incubation in the dark at 37 °C, cells were washed twice with PBS and harvested. Samples were acquired on a flow cytometer (Navios EX, Beckman Coulter, Pasadena, CA, USA) and results were analysed using Kaluza Flow Cytometry Analysis v2.1 software (Beckman Coulter, Pasadena, CA, USA). Relative fluorescence intensities were obtained from the median fluorescence intensity of each histogram of the different experimental conditions.

### 4.10. DNA Damage Detection

DNA damage was assessed using the Muse^®^ Multi-Color DNA Damage kit (Luminex), allowing to distinguish four distinct populations: percentage of negative cells, pATM+ cells, γH2AX+ cells and both pATM and γH2AX+ cells. Briefly, supernatant and adherent cells of four wells were harvested, pooled and re-suspended in 1x assay buffer. Cells were fixed using the fixation buffer, permeabilized using the permeabilization buffer, resuspended in 1x assay buffer and then stained with anti-phospho-ATM (Ser1981)-PE and anti-phospho-Histone H2AX (Ser139)-PECy5-conjugated antibodies for 30 min in the dark at room temperature. Every sample was then thoroughly mixed and run on the Muse^®^ Cell Analyzer (Merck). The analysis was performed on 2000 cells to reach the statistical threshold [45].

### 4.11. Statistical Analyses

Statistical analyses were performed using the GraphPad Prism 7 software. After normality verification using the Shapiro–Wilk test, parametric t-test was performed. Two-way ANOVA with a Turkey’s multiple comparison test was used to assess differences in the cell cycle distribution. Significance for t-test results between treated cells and non-treated cells is indicated as: * *p* ≤ 0.05, ** *p* ≤ 0.01 and *** *p* ≤ 0.001. A different symbol (#) is used to represent significant differences between ^177^Lu-DOTATATE- and EBRT-treated cells.

## Figures and Tables

**Figure 1 ijms-23-12369-f001:**
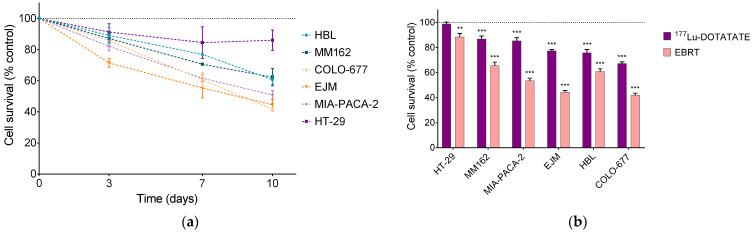
Effect of ^177^Lu-DOTATATE and EBRT on the survival of melanoma (HBL and MM162), multiple myeloma (COLO-677 and EJM) and GEP (MIA-PACA-2 and HT-29) cell lines. (**a**) Cell survival was assessed 3, 7 and 10 days after 2 Gy EBRT. (**b**) Cell survival at day 10 after 4 h incubation with 5 MBq of ^177^Lu-DOTATATE (purple) or 2 Gy EBRT (pink). Results are expressed as a percentage of the non-treated counterpart (black dotted line) and are represented as mean ± SEM (*n* = 20 from 5 independent experiments): ** *p* ≤ 0.01; *** *p* ≤ 0.001.

**Figure 2 ijms-23-12369-f002:**
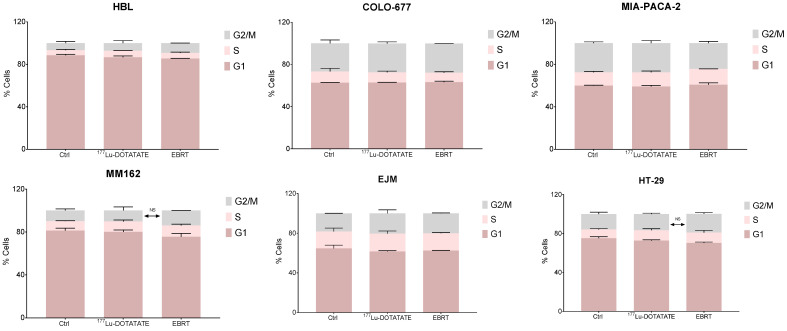
Effect of ^177^Lu-DOTATATE and EBRT on the cell cycle distribution in melanoma (HBL and MM162), multiple myeloma (COLO-677 and EJM) and GEP (MIA-PACA-2 and HT-29) cell lines. Cell cycle was assessed 10 days after 4 h incubation with 5 MBq of ^177^Lu-DOTATATE or 2 Gy EBRT. Results are represented as mean ± SEM (*n* = 3 from 3 independent experiments).

**Figure 3 ijms-23-12369-f003:**
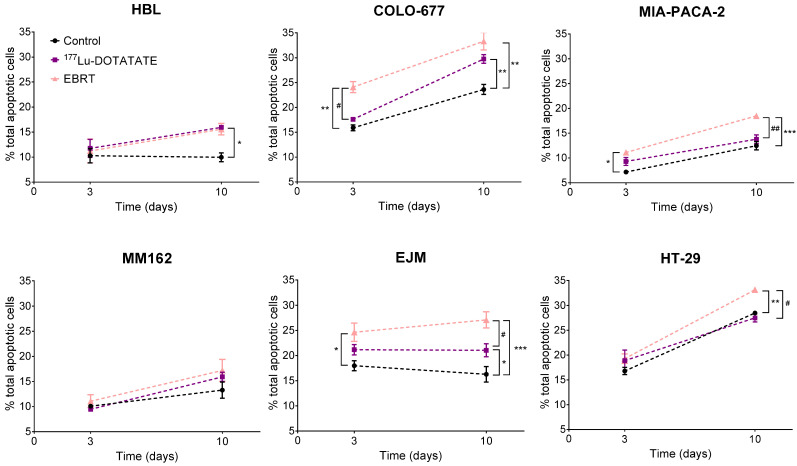
Effect of ^177^Lu-DOTATATE and EBRT on apoptosis induction in melanoma (HBL and MM162), multiple myeloma (COLO-677 and EJM) and GEP (MIA-PACA-2 and HT-29) cell lines. The percentage of total apoptotic cells (Annexin V+ 7-AAD-, Annexin V+ 7-AAD+) was assessed 3 and 10 days after 4 h incubation with 5 MBq of ^177^Lu-DOTATATE (purple) or 2 Gy EBRT (pink). Results are represented as mean ± SEM (*n* = 3 from 3 independent experiments). Only the significant differences are indicated on the graphs: * *p* ≤ 0.05; ** *p* ≤ 0.01; *** *p* ≤ 0.001. The following symbols were used for the comparison between ^177^Lu-DOTATATE and EBRT results: # *p* ≤ 0.05; ## *p* < 0.01.

**Figure 4 ijms-23-12369-f004:**
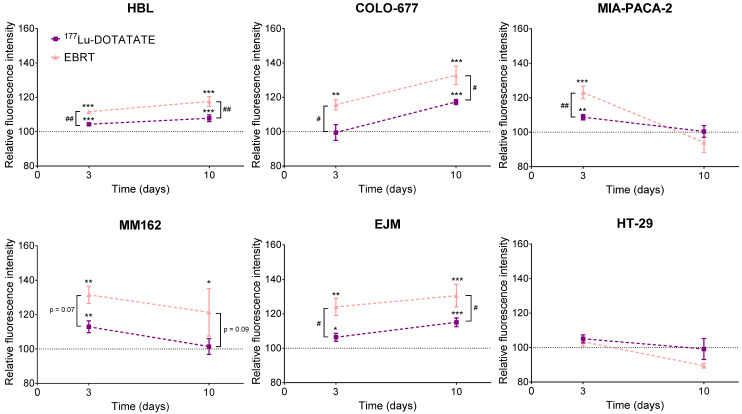
Effect of ^177^Lu-DOTATATE and EBRT on autophagy induction in melanoma (HBL and MM162), multiple myeloma (COLO-677 and EJM) and GEP (MIA-PACA-2 and HT-29) cell lines. Autophagy was assessed 3 and 10 days after 4 h incubation with 5 MBq of ^177^Lu-DOTATATE (purple) or 2 Gy EBRT (pink). Results are expressed as a percentage of the non-treated counterpart (black dotted line) and are represented as mean ± SEM (*n* = 3 from 3 independent experiments): * *p* ≤ 0.05; ** *p* ≤ 0.01; *** *p* ≤ 0.001. The following symbols were used for the comparison between ^177^Lu-DOTATATE and EBRT results: # *p* ≤ 0.05; ## *p* < 0.01.

**Figure 5 ijms-23-12369-f005:**
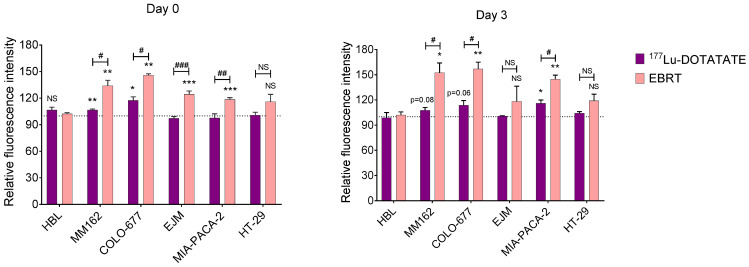
Effect of ^177^Lu-DOTATATE and EBRT on ROS induction in melanoma (HBL and MM162), multiple myeloma (COLO-677 and EJM) and GEP (MIA-PACA-2 and HT-29) cell lines. ROS were assessed immediately (within 15 min) after 4 h incubation with 5 MBq of ^177^Lu-DOTATATE (purple) or 2 Gy EBRT (pink) and at day 3. Results are normalized against the non-treated counterpart (black dotted line) (*n* = 6 from 3 independent experiments). NS: not statistically significant; * *p* ≤ 0.05; ** *p* ≤ 0.01; *** *p* ≤ 0.001. The following symbols were used for the comparison between ^177^Lu-DOTATATE and EBRT results: # *p* ≤ 0.05; ## *p* < 0.01; ### *p* < 0.001.

**Figure 6 ijms-23-12369-f006:**
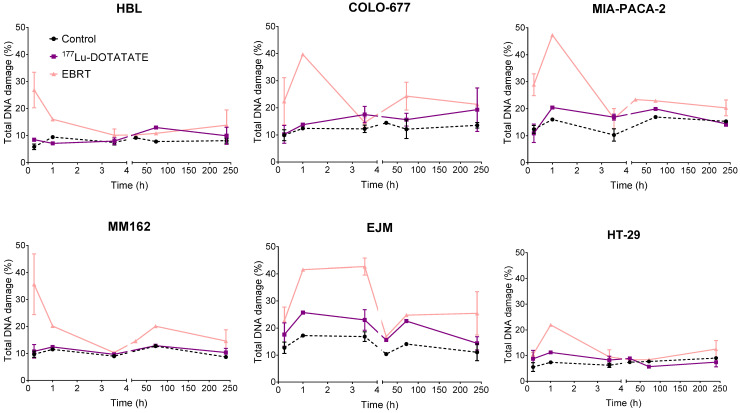
Effect of ^177^Lu-DOTATATE and EBRT on DNA damage repair in melanoma (HBL and MM162), multiple myeloma (COLO-677 and EJM) and GEP (MIA-PACA-2 and HT-29) cell lines. Total DNA damage (pATM+ γH2AX-, pATM- γH2AX+, pATM+ γH2AX+) was assessed at different timepoints after 4 h incubation with 5 MBq of ^177^Lu-DOTATATE (purple) or 2 Gy EBRT (pink). Results are represented as mean ± SEM (*n* = 3 from 3 independent experiments for 15 min, 3.5 h and 10-day time points).

**Figure 7 ijms-23-12369-f007:**
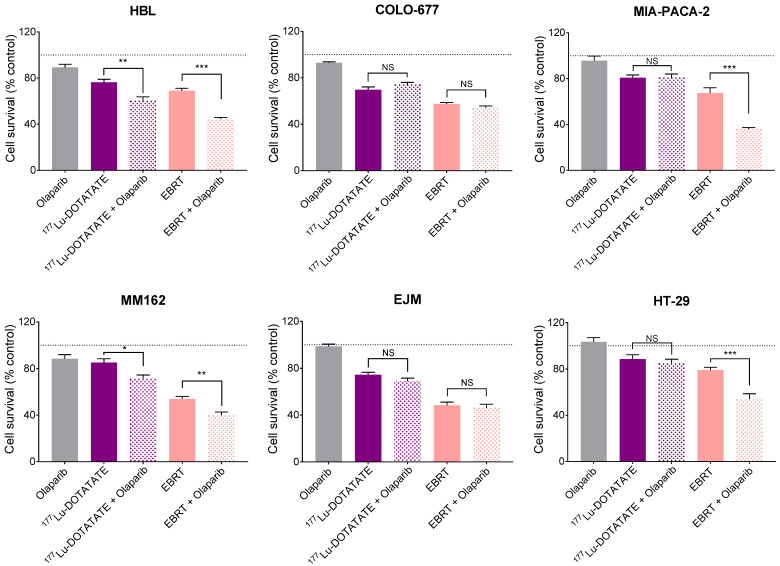
Effect of olaparib and its combination with ^177^Lu-DOTATATE and EBRT on survival of melanoma (HBL and MM162), multiple myeloma (COLO-677 and EJM) and GEP (MIA-PACA-2 and HT-29) cell lines. Cell survival was assessed 10 days after 4 h incubation with 5 MBq of ^177^Lu-DOTATATE (purple) or 2 Gy EBRT (pink). Olaparib (HBL, MM162 and HT-29: 10^−6^ M; MIA-PACA-2 and EJM: 10^−7^ M; COLO-677: 10^−8^ M) was present in the medium from the day before irradiation until the end of the experiment. Results are expressed as a percentage of the non-treated counterpart (black dotted line) and are represented as mean ± SEM (*n* = 12 from 3 independent experiments): *** *p* ≤ 0.001; ** *p* ≤ 0.01; * *p* ≤ 0.05; NS = not statistically significant.

**Figure 8 ijms-23-12369-f008:**
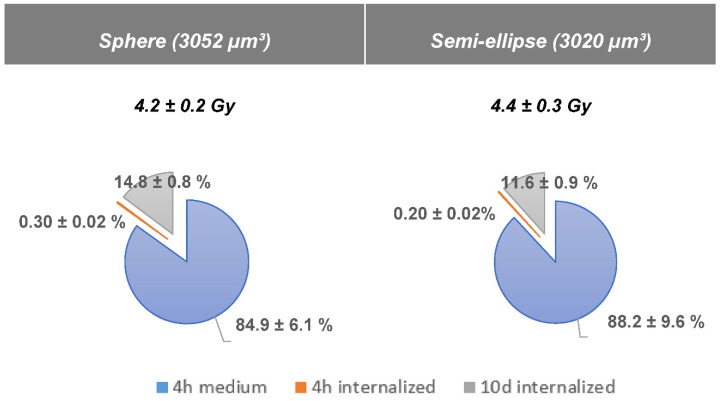
Mean cumulative absorbed dose ± SD to cell for 2 different geometries with a constant volume (sphere and semi-ellipse of 3052 and 3020 µm^3^ respectively) with contribution in % of the medium (in blue), the internalized fraction during the 4h (in orange) and the 10 days incubation (in grey).

**Figure 9 ijms-23-12369-f009:**
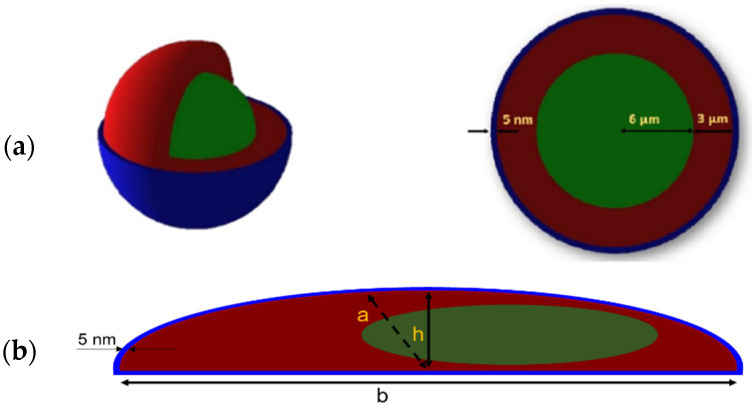
(**a**) Spherical and (**b**) semi-ellipsoid cell models used for the simulation set-up (nucleus in green, cytoplasm in red, membrane in blue): a = minor axis; b = major axis; h = height. Ratios are as follows: a/b = 1/3; h/a = 1/2.

**Table 1 ijms-23-12369-t001:** The coefficient of drug interaction (CDI) and amplification factor (AF) for ^177^Lu-DOTATATE and EBRT combined with olaparib at minimal toxic concentration. CDI = survival%(A + B)/(survival%(A) × survival%(B)), where A is either ^177^Lu-DOTATATE or EBRT and B is olaparib. CDI  <  1 indicates synergism, CDI  =  1 additivity and CD  >  1 antagonism. AF (%) = (survival%(A)-survival%(A + B))/survival%(A) × 100, where A is either ^177^Lu-DOTATATE or EBRT and B is olaparib. NA = not applicable.

	CDI		AF (%)	
	^177^Lu-DOTATATE	EBRT	^177^Lu-DOTATATE	EBRT
HBL	0.84	0.72	20	34
MM162	0.95	0.80	12	26
COLO-677	NA	NA	0	5
EJM	NA	NA	7	5
MIA-PACA-2	NA	0.58	0	43
HT-29	NA	0.63	4	34

**Table 2 ijms-23-12369-t002:** Sources of electrons and photons used in the simulations.

	% of Branching Ratios Used for the Simulations [44]	Energies (keV)
Gamma	17.3	71, 113, 136, 208, 250 and 321
X-rays	8.5	9, 55, 56 and 64
β particles	100.0	0 to 497 (E_max_)
Auger electrons	22.1	8, 48, 103 and 111

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
