# Peer review of "Understanding the Radiobiological Mechanisms Induced by 177Lu-DOTATATE in Comparison to External Beam Radiation Therapy"

_ijms, 2022, doi:10.3390/ijms232012369_

Round 1
Reviewer 1 Report
Introduction is well written, and the author has the right emphasis on the lack of knowledge for the RNT radiobiology. However, I have a minor comment about the beginning of the introduction: It is better to say the “targeted RNT” is an effective systemic therapy. There are different RNT modalities that are not systemic such as 90Y-SIRT.
Results:
Although the authors have provided number of analyses to benchmark the RNT and EBRT, I would like to have clarification on the method section. The comments and questions below are critical to understand whether the experiments were performed correctly.
1. Could the author provide the alpha and beta parameters for the 6 cell lines. These radiobiological parameters are critical for the design of the 177Lu treatment of the cells. Could the author also calculate the initial dose rate at which the cells were treated with 177 Lu (both with Cl3 and DOTATATE cases). The cell dosimetry was considered 10 days incubation of the 177Lu, however, I am not sure if the dose rate was dropped below the critical dose rate at sometime during the cell treatment. Any absorbed dose beyond the critical dose rate is wasted. So, it is important to choose an initial dose rate that does not drops beyond the critical dose rate during the incubation time. Otherwise, the cell dosimetry is not accurate.
2. In this study the 5 MBq of 177Lu (both with Cl3 and DOTATATE form) was compared against the 2 Gy acute dose of EBRT. However, the dose rate at which the dose is delivered to the cells are different (i.e., the EBRT is acute with G factor of ≈1 and 177Lu is exponentially decreasing with a G factor of much less than 1). For benchmarking, similar dose rate of different type of radiation is required. I think the author might need to include an additional set of experiment including the fractionated EBRT cell treatment.
3. I think it make more sense to include the BED calculation to this work since this study is focusing on the understanding the radiobiological mechanisms induced by RNT and EBRT.
4. For MC simulation, are the cells overlapping? What algorithm has the author used to create the cell cluster?
5. For MC simulation, the branching ratios are incorrect (e.g., the beta- particle branching ratio should be ≈ 79%), please check the IAEA nuclear data (see the link below). The simulation work needs to be repeated by the correct branching ratio.
https://www-nds.iaea.org/relnsd/vcharthtml/VChartHTML.html
Also for figure 1 (b), 3, 4,6, there are some word overlapped with the y-axis title.
Reviewer 2 Report
see attached file

Round 2
Reviewer 1 Report
None.